# Learning Sparse Prototypes for Text Generation

**Junxian He[†], Taylor Berg-Kirkpatrick[‡], Graham Neubig[†]**
[†]Carnegie Mellon University; [‡]University of California San Diego
{junxianh,gneubig}@cs.cmu.edu, tberg@eng.ucsd.edu

## Abstract

Prototype-driven text generation uses non-parametric models that first choose from a library of sentence "prototypes" and then modify the prototype to generate the output text. While effective, these methods are inefficient at test time as a result of needing to store and index the entire training corpus. Further, existing methods often require heuristics to identify which prototypes to reference at training time. In this paper, we propose a novel generative model that automatically learns a *sparse* prototype support set that, nonetheless, achieves strong language modeling performance. This is achieved by (1) imposing a sparsity-inducing prior on the prototype selection distribution, and (2) utilizing amortized variational inference to *learn* a prototype retrieval function. In experiments, our model outperforms previous prototype-driven language models while achieving up to a 1000x memory reduction, as well as a 1000x speed-up at test time. More interestingly, we show that the learned prototypes are able to capture semantics and syntax at different granularity as we vary the sparsity of prototype selection, and that certain sentence attributes can be controlled by specifying the prototype for generation.[1]

## 1 Introduction

Language models (LMs) predict a probability distribution over text, and are a fundamental technology widely studied in the natural language processing (NLP) community (Bengio et al., 2003; Merity et al., 2018; Dai et al., 2019). Modern LMs are almost exclusively based on *parametric* recurrent (Mikolov et al., 2010; Sundermeyer et al., 2012) or self-attentional (Vaswani et al., 2017; Al-Rfou et al., 2019) neural networks. These models are of interest scientifically as one of the purest tests of our ability to capture the intricacies of human language mathematically (Linzen et al., 2016; Kuncoro et al., 2017; Petroni et al., 2019). They also have broad downstream applications in generating text in systems such as machine translation (Bahdanau et al., 2015), summarization (Rush et al., 2015), or dialog generation (Sordoni et al., 2015), as well as in the unsupervised representation learners that now power many applications in NLP (Devlin et al., 2018; Liu et al., 2019; Yang et al., 2019).

However, there has been a recent move towards *non-parametric* neural LMs (Guu et al., 2018; Khandelwal et al., 2020b) that generate sentences by first selecting examples from an external datastore. For instance, Khandelwal et al. (2020b) model the token-level probability at test time by interpolating the language model with a kNN distribution from the nearest context-token pairs in the datastore, while Guu et al. (2018) store external memories on sentence level and feature a prototype-then-edit process of (1) selecting a *prototype* sentence from a the prototype datastore, and (2) editing this prototype to the final desired output. In this paper, we focus on the prototype-then-edit model family which is a lot lighter relatively in terms of memory and time cost at test time.

Intuitively, these non-parametric LMs are attractive because they help remove some of the pressure on the parametric model to memorize the entirety of the language it must model. These intuitive advantages are also borne out in superior performance on language modeling tasks (Guu et al.,

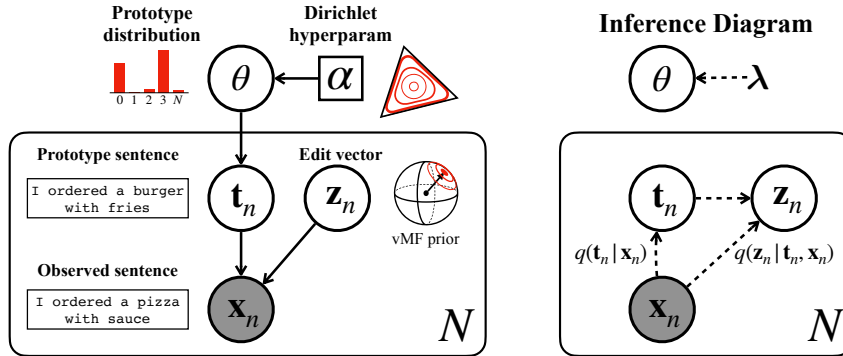

Figure 1: **Left:** the proposed generative model to generate data by editing prototypes. Shaded circles denote the observed variables and unshaded denote the latents. Prototypes are sampled from a sparse prototype distribution which itself is a random variable sampled from a Dirichlet prior distribution. **Right:** the inference diagram of the model, with $q(\mathbf{t}_n|\mathbf{x}_n)$ being the prototype retriever and $q(\mathbf{z}_n|\mathbf{t}_n, \mathbf{x}_n)$ being the inverse editor.

2018; Khandelwal et al., 2020b), as well as down-stream applications such as dialogue response generation (Weston et al., 2018; Wu et al., 2019), machine translation (Gu et al., 2018; Bapna & Firat, 2019; Khandelwal et al., 2020a), and code generation (Hashimoto et al., 2018; Hayati et al., 2018). In addition, the prototypes and continuous representations of the edits in prototype-based models lend an element of interpretability to the modeling process. On the down side, however, previous prototype-driven generation methods usually need to store and index a large prototype candidate pool (in general the whole training dataset), leading to significant issues with memory and speed efficiency at test time.

In this paper, we hypothesize that, in fact, a *small* set of prototypes is sufficient to achieve the great majority of the gains afforded by such non-parametric models. Intuitively, in a large corpus many sentences look very similar and may be represented by minor transformations of a single prototype sentence. For example, the sentence "I ordered a burger with fries" can serve as the prototype for data samples with the form "I ordered [NOUN PHRASE] with [NOUN PHRASE]". This is evidenced by Guu et al. (2018)'s observation that 70% of the test set in the Yelp restaurant review corpus (Yelp, 2017) is within word-token Jaccard distance 0.5 of one training sentence.

To take advantage of this intuition, we propose a novel generative model that samples prototypes from a *latent* prototype distribution, which itself is sampled from a symmetric Dirichlet prior, as shown in Figure 1 (Section 3.1). The Dirichlet prior with appropriate hyperparameters is able to encourage a *sparse* prototype selection distribution, allowing us to reduce the prototype support set at test time to greatly improve efficiency. Moreover, we utilize amortized variational inference (Kingma & Welling, 2013) to train our model, which introduces a learnable *prototype retriever* to identify prototypes useful for generating each sentence (Section 3.2). This is different from (Guu et al., 2018) where prototypes for each sentence are fixed before training through edit distance heuristics.

We evaluate our approach on the MSCOCO (Lin et al., 2014) and Yelp restaurant review (Yelp, 2017) corpora. Our method is able to improve perplexity over the neural language model baseline by up to 14 points and previous neural editor model by 6 points while achieving over 1000x memory savings and a 1000x speedup at test time (Section 4.2). Interestingly, we find that the learned prototypes are able to represent different features when varying sparsity levels – a strong sparsity prior forces the model to share prototypes and the induced prototypes turn out to represent more generic features (e.g. syntactic form of the sentence). On the text generation side, our model is able to generate sentences that resemble the given prototype while allowing for smooth interpolation on the edit space as well (Section 4.3).

## 2 Background

The prototype-then-edit framework defines a non-parametric way to augment text generation models. Formally, given a corpus $\mathbf{X} = \{\mathbf{x}_n\}_{n=1}^{N}$,[2] the model generates each observed example $\mathbf{x}_n$ by: (1) retrieving a prototype sequence $\mathbf{t}_n$, (2) generating a continuous edit representation $\mathbf{z}_n$, and (3)

generating $\mathbf{x}_n$ conditioned on $\mathbf{t}_n$ and $\mathbf{z}_n$. These intermediate prototype and edit representation variables can depend on extra context in conditional generation tasks (Hodosh et al., 2013; Gu et al., 2018), or are randomly sampled in unconditioned language modeling. In this paper, we focus on the latter, but our methods could likely be applied to the former as well.

For unconditioned language modeling, Guu et al. (2018) define the data likelihood as:

$$p(\mathbf{X}) = \prod_n \sum_{\mathbf{t}_n} \int_{\mathbf{z}_n} p(\mathbf{z}_n)p(\mathbf{t}_n)p_{\boldsymbol{\gamma}}(\mathbf{x}_n|\mathbf{t}_n, \mathbf{z}_n)d\mathbf{z}_n, \tag{1}$$

where $p(\mathbf{t}_n)$ is the prior distribution over prototypes and defined as a uniform distribution over *all* training examples, $p(\mathbf{z}_n)$ is a continuous distribution over the edit vector, and $p_{\boldsymbol{\gamma}}(\mathbf{x}_n|\mathbf{t}_n, \mathbf{z}_n)$ represents a sequence-to-sequence model parameterized by $\boldsymbol{\gamma}$. This model is referred to as the *neural editor*. Guu et al. (2018)'s stated goal of the neural editor is to take direct advantage of training examples to improve language modeling performance while capturing interpretable semantic or syntactic properties in the latent prototype and edit vector variables. However, because the prototypes are selected from the *entire* training dataset, such a formulation sacrifices memory and speed efficiency due to the necessity of indexing and searching every training example at test time. In the following section, we detail our approach to mitigate this issue through the learning of *sparse* prototypes.

## 3 Method

First we present our our proposed generative model, then we describe the learning and inference techniques for this model class.

### 3.1 Model Structure

In the previous formulation, Eq. 1, maintaining the entire training dataset at test time is necessary due to assuming a uniform prior over prototypes $p(\mathbf{t})$. Motivated by the hypothesis that a much smaller prototype set would suffice to achieve comparable performance, however, we believe that $p(\mathbf{t})$ can be a sparse distribution where the probability mass concentrates on only a few representative prototypes. Since which training examples are representative as prototypes is unknown in advance, we propose to model the prototype distribution $p(\mathbf{t})$ as a latent variable, endowing the model with freedom to infer a sparse prototype posterior automatically. We define $\theta \equiv p(\mathbf{t})$ and further assume that the latent prototype distribution $\theta$ is sampled from a prior distribution $p_\alpha(\theta)$ (detailed below) which is able to encourage a sparse probability distribution, given appropriate hyperparameters. The graphical model is depicted in Figure 1, which gives the following joint likelihood:

$$p(\{\mathbf{x}_n, \mathbf{t}_n, \mathbf{z}_n\}_{n=1}^N, \theta; \alpha, \boldsymbol{\gamma}) = p_\alpha(\theta) \prod_n p(\mathbf{t}_n|\theta)p(\mathbf{z}_n)p_{\boldsymbol{\gamma}}(\mathbf{x}_n|\mathbf{t}_n, \mathbf{z}_n). \tag{2}$$

The log marginal likelihood of the data, which we will approximate during training is:

$$\log p(\mathbf{X}; \boldsymbol{\gamma}, \alpha) = \log \int_\theta p_\alpha(\theta) \Big[ \prod_n \sum_{\mathbf{t}_n} \int_{\mathbf{z}_n} p(\mathbf{t}_n|\theta)p(\mathbf{z}_n)p_{\boldsymbol{\gamma}}(\mathbf{x}_n|\mathbf{t}_n, \mathbf{z}_n)d\mathbf{z}_n \Big] d\theta. \tag{3}$$

This is a general framework for learning sparse prototypes that we refer to as the *sparse neural editor*, and in this work we specifically experiment with the following parameterization to instantiate this model class:

**Prior over prototype distribution $p_\alpha(\theta)$:** We employ the Dirichlet distribution as $p_\alpha(\theta)$: $p_\alpha(\theta) \propto \prod_{k=1}^N \theta_k^{\alpha_k - 1}$. The support of Dirichlet distribution is the standard $N-1$ probability simplex. Here we use the *symmetric* Dirichlet distribution which has the same $\alpha$ value for all components since we have no prior knowledge favoring one component over another. $\alpha$ is positive and also referred to as the concentration parameter, where smaller $\alpha$ prefers a sparser prototype distribution $\theta$, with $\alpha = 1$ equivalent to a uniform distribution over the probability simplex. In our experiments, we often choose $\alpha < 1$ to encourage sparsity.

**Prior over edit vector $p(\mathbf{z})$:** We follow Guu et al. (2018) and utilize a von-Mises Fisher (vMF) distribution to model $p(\mathbf{z})$. The vMF distribution places mass on the surface of the unit sphere, and is parameterized by the mean direction vector $\boldsymbol{\mu}$ and concentration parameter $\kappa$ as vMF$(\cdot|\boldsymbol{\mu}, \kappa)$. Thus,

information about the edit is captured through the directions of different unit vector samples. Xu & Durrett (2018) empirically show that the vMF distribution has the advantage of overcoming posterior collapse that plagues a large amount of previous work in latent variable models of text (Bowman et al., 2016; He et al., 2019). While Guu et al. (2018) add additional randomness on the norm of edit vectors by multiplying the vMF distribution with another uniform distribution, we sample edit vectors from a uniform vMF distribution directly, which simplifies the model but we nonetheless found sufficient to obtain competitive results. Formally, we define $p(\mathbf{z}) = \text{vMF}(\mathbf{z}|\cdot, 0)$.

**The editor** $p_{\boldsymbol{\gamma}}(\mathbf{x}|\mathbf{t}, \mathbf{z})$**:** Generally $p_{\boldsymbol{\gamma}}(\mathbf{x}|\mathbf{t}, \mathbf{z})$ can be parameterized by any standard Seq2Seq model with the edit vector $\mathbf{z}$ incorporated. To compare with Guu et al. (2018) directly in the experiments, in this work we adopt the same attentional LSTM architecture (Hochreiter & Schmidhuber, 1997). $\mathbf{z}$ is utilized to predict the initial hidden state of the decoder and concatenated to the input for the decoder.

### 3.2 Learning and Inference

Ideally the log marginal likelihood in Eq. 3 should be optimized during training. However, computation is intractable due to marginalization of latent variables, and we resort to amortized variational inference (Kingma & Welling, 2013), optimizing its evidence lower bound (ELBO) instead:

$$
\begin{aligned}
\log p(\mathbf{X}; \boldsymbol{\gamma}, \alpha) &\geq \mathcal{L}_{\text{ELBO}}(\mathbf{X}; \boldsymbol{\gamma}, \alpha, \boldsymbol{\phi}_{t|x}, \boldsymbol{\phi}_{z|t,x}, \boldsymbol{\lambda}) \\
&= \sum_n \Bigg\{ \underbrace{\mathbb{E}_{q_{\boldsymbol{\phi}_{t|x}}(\mathbf{t}_n|\mathbf{x}_n) q_{\boldsymbol{\phi}_{z|t,x}}(\mathbf{z}_n|\mathbf{t}_n, \mathbf{x}_n)}[\log p_{\boldsymbol{\gamma}}(\mathbf{x}_n|\mathbf{t}_n, \mathbf{z}_n)]}_{\text{reconstruction log likelihood } \mathcal{L}_{\text{rec}}} \\
&\quad - \mathbb{E}_{q_{\boldsymbol{\phi}_{t|x}}(\mathbf{t}_n|\mathbf{x}_n)}[D_{\text{KL}}(q_{\boldsymbol{\phi}_{z|t,x}}(\mathbf{z}_n|\mathbf{t}_n, \mathbf{x}_n)||p(\mathbf{z}_n))] \\
&\quad - \mathbb{E}_{q_{\boldsymbol{\lambda}}(\theta)}[D_{\text{KL}}(q_{\boldsymbol{\phi}_{t|x}}(\mathbf{t}_n|\mathbf{x}_n)||p(\mathbf{t}_n|\theta))] \Bigg\} - D_{\text{KL}}(q_{\boldsymbol{\lambda}}(\theta)||p_{\alpha}(\theta)),
\end{aligned}
\tag{4}
$$

where $q$ represents the variational distribution to approximate the model posterior distribution and admits the following factorization form:

$$
q(\theta, \{\mathbf{t}_n, \mathbf{z}_n\}_{n=1}^N|\mathbf{X}; \boldsymbol{\lambda}, \boldsymbol{\phi}_{t|x}, \boldsymbol{\phi}_{z|t,x}) = q_{\boldsymbol{\lambda}}(\theta) \prod_n q_{\boldsymbol{\phi}_{t|x}}(\mathbf{t}_n|\mathbf{x}_n) q_{\boldsymbol{\phi}_{z|t,x}}(\mathbf{z}_n|\mathbf{t}_n, \mathbf{x}_n).
\tag{5}
$$

Note that we make conditional independence assumption between $\theta$ and other latent variables in $q$ to simplify the approximate posterior, following common practice in traditional mean field variational inference. The inference diagram is depicted in Figure 1. The optimal $q_{\boldsymbol{\lambda}}(\theta)$ to maximize Eq. 4 is a Dirichlet distribution parameterized by $\boldsymbol{\lambda} \in \mathbb{R}_+^N$ (proof is in Appendix A), i.e., $q_{\boldsymbol{\lambda}}(\theta) = \text{Dir}(\theta; \boldsymbol{\lambda})$.[3] And $q_{\boldsymbol{\phi}_{t|x}}(\mathbf{t}|\mathbf{x}) = \text{Cat}(\mathbf{t}; f_{\boldsymbol{\phi}_{t|x}}(\mathbf{x}))$, the *prototype retriever*, is a categorical distribution over training examples parameterized by a neural network $f_{\boldsymbol{\phi}_{t|x}}(\mathbf{x})$. We assume $q_{\boldsymbol{\phi}_{z|t,x}}(\mathbf{z}|\mathbf{t}, \mathbf{x})$, the *inverse neural editor*, is a vMF distribution $q_{\boldsymbol{\phi}_{z|t,x}}(\mathbf{z}|\mathbf{t}, \mathbf{x}) = \text{vMF}(g_{\boldsymbol{\phi}_{z|t,x}}(\mathbf{t}, \mathbf{x}), \kappa)$ where the mean direction parameter is from an encoder $g$ that encodes $\mathbf{t}$ and $\mathbf{x}$ parameterized by $\boldsymbol{\phi}_{z|t,x}$, and the scalar concentration parameter $\kappa$ is a hyperparameter. Pre-fixing $\kappa$ results in a constant KL divergence term associated with $\mathbf{z}$ and proves to be effective to mitigate the posterior collapse issue (Xu & Durrett, 2018) where $\mathbf{x}$ and $\mathbf{z}$ become independent. Yet there might be still posterior collapse on $\mathbf{t}$ where, for example, the prototype retriever always predicts a uniform distribution or a degenerate distribution concentrated on a certain prototype regardless of $\mathbf{x}$. To overcome this issue, we follow (Li et al., 2019) to combine annealing (Bowman et al., 2016) and free-bits techniques (Kingma et al., 2016) and apply them to the term $\mathbb{E}_{q_{\boldsymbol{\lambda}}(\theta)}[D_{\text{KL}}(q_{\boldsymbol{\phi}_{t|x}}(\mathbf{t}_n|\mathbf{x}_n)||p(\mathbf{t}_n|\theta))]$.

Notably, the variational distribution family defined in Eq. 5 admits tractable closed-form expressions of all three KL divergence terms in Eq. 4 (detailed derivations and expressions are in Appendix A). To compute the reconstruction log likelihood $\mathcal{L}_{\text{rec}}$, expectations over $\mathbf{z}$ can be efficiently approximated by the reparameterization trick for the vMF distribution (Guu et al., 2018). However, the prototype $\mathbf{t}$ is discrete and non-differentiable, and summing over all prototypes $\mathbf{t}$ to compute $\mathcal{L}_{\text{rec}}$ is infeasible due to the evaluation burden of $\log p_{\boldsymbol{\gamma}}(\mathbf{x}|\mathbf{t}, \mathbf{z})$. Thus, we use the REINFORCE algorithm (Williams, 1992) to compute the gradients of $\boldsymbol{\phi}_{t|x}$ contributed from $\mathcal{L}_{\text{rec}}$ as:

$$
\frac{\partial \mathcal{L}_{\text{rec}}}{\partial \boldsymbol{\phi}_{t|x}} = \frac{1}{L} \sum_{l=1}^L \Big( \mathbb{E}_{q_{\boldsymbol{\phi}_{z|t,x}}(\mathbf{z}|\mathbf{t}^{(l)}, \mathbf{x})}[\log p_{\boldsymbol{\gamma}}(\mathbf{x}|\mathbf{t}^{(l)}, \mathbf{z})] - b \Big) \frac{\partial \log q_{\boldsymbol{\phi}_{t|x}}(\mathbf{t}^{(l)}|\mathbf{x})}{\partial \boldsymbol{\phi}_{t|x}},
\tag{6}
$$

where $\mathbf{t}^{(l)}$ are samples from $q_{\phi_{t|x}}(\mathbf{t}|\mathbf{x})$. We use an average reward from $L$ samples as the baseline $b$. The neural parameters $\gamma, \phi_{t|x}, \phi_{z|t,x}$ are updated with stochastic gradient descent to maximize Eq. 4. With respect to the posterior Dirichlet parameter $\boldsymbol{\lambda}$, we found in preliminary experiments that classic gradient descent was unable to update it effectively – $\boldsymbol{\lambda}$ was updated too slowly and the Dirichlet prior became decoupled with the model. Thus, we instead update $\boldsymbol{\lambda}$ with stochastic variational inference (SVI, Hoffman et al. (2013)) based on the formula of the optimal $\boldsymbol{\lambda}^*$ given $q_{\phi_{t|x}}(\mathbf{t}|\mathbf{x})$ (derivations can be found in Appendix A):

$$\lambda_k^* = \alpha + \sum\nolimits_{n=1}^{N} q_{\phi_{t|x}}(\mathbf{t}_n = \mathbf{x}_k|\mathbf{x}_n). \tag{7}$$

It is infeasibly expensive to keep $\boldsymbol{\lambda}$ optimal under current $q_{\phi_{t|x}}(\mathbf{t}|\mathbf{x})$ at each training step, as it would involve summing over all training examples. Thus we perform SVI, which uses a batch of examples to approximate Eq. 7, leading to the following update form:

$$\lambda_k^{(t)} = (1 - \rho_t)\lambda_k^{(t-1)} + \rho_t\Big(\alpha + \frac{N}{B}\sum\nolimits_{i=1}^{B} q_{\phi_{t|x}}(\mathbf{t}_i = \mathbf{x}_k|\mathbf{x}_i)\Big), \quad \rho_t = (t + \sigma)^{-\tau}, \tag{8}$$

where $B$ is the batch size, $\rho_t$ is the step-size at iteration $t$, $\tau \in (0.5, 1]$ is the forgetting rate, and $\sigma \geq 0$ is the delay parameter to down-weight early iterations.

We note that our training algorithm is different from Guu et al. (2018) in that we use a learnable prototype retriever $q_{\phi_{t|x}}(\mathbf{t}|\mathbf{x})$ to derive a lower bound as the objective while Guu et al. (2018) directly approximate marginalization over $\mathbf{t}$. They use heuristics to fix the prototype set for each $\mathbf{x}$ to be examples similar to $\mathbf{x}$ in terms of edit distance, which might produce suboptimal prototypes for the generative model and also does not permit the learning of sparse prototype support.

**Sparsity and scalability:** After training we expect to be able to infer a *sparse* prototype distribution $\theta$ with most components being almost zero, based on which we can prune and store the entries over a particular probability threshold only, improving memory- and time-efficiency at test time. Specifically, we compute mean of $\theta$ under the Dirichlet posterior: $\mathbb{E}_{q_{\boldsymbol{\lambda}}(\theta)}[\theta_k] = \lambda_k / \sum_i \lambda_i$, and then take the largest $M$ entries that occupy $90\%$ of the probability mass. At test time, we only maintain these $M$ prototypes and the prototype retriever $q_{\phi_{t|x}}(\mathbf{t}|\mathbf{x})$ is re-normalized accordingly. One issue present during training is that $q_{\phi_{t|x}}(\mathbf{t}|\mathbf{x})$ cannot fit into memory when dealing with large datasets since it is a categorical distribution over all training examples. In this work, we randomly downsample a subset of training data as our prototype library before training if memory is unable to fit all training examples, and learn the sparse prototypes on top of this subsampled corpus. This acts like a rough pre-filtering and in Section 4 we show that it suffices to learn good prototypes and achieve competitive language modeling performance. We leave more advanced techniques to address this issue (e.g. dynamically updating the prototype library) as future work.

**Architectures:** We now describe the neural architectures we use for the prototype retriever $q(\mathbf{t}|\mathbf{x})$ and inverse neural editor $q(\mathbf{z}|\mathbf{t}, \mathbf{x})$. $q(\mathbf{t}|\mathbf{x})$ is defined as:

$$q(\mathbf{t} = \mathbf{x}_k|\mathbf{x}) \propto \begin{cases} \exp\big(h(\mathbf{x}_k, \mathbf{x})/\mu\big) & \text{if } \mathbf{x}_k \neq \mathbf{x} \\ 0 & \text{if } \mathbf{x}_k = \mathbf{x}, \end{cases} \quad h(\mathbf{x}_k, \mathbf{x}) = \texttt{Embed}(\mathbf{x}_k)^\top \boldsymbol{W} \texttt{Embed}(\mathbf{x}), \tag{9}$$

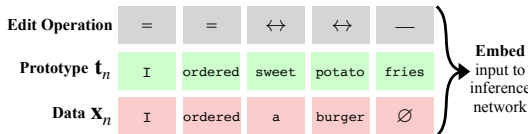

Figure 2: Example of aligned sequences.

where we prevent selecting the data example itself as the prototype during training to avoid overfitting. $\texttt{Embed}(\cdot)$ is a pretrained sentence encoder, $\boldsymbol{W}$ is a linear transformation matrix, and $\mu$ is a temperature hyperparameter to control the entropy of $q_{\phi_{t|x}}(\mathbf{t}|\mathbf{x})$, which is critical to stabilize the Reinforce algorithm at the initial training stage. To ease the computation of encoding all training examples at each update step, we fix the parameters of $\texttt{Embed}(\cdot)$ and update $\boldsymbol{W}$ only, which proves to be sufficient in our experiments. We use the average embeddings of the last layer in pretrained BERT (Devlin et al., 2018)[4] as our sentence encoder.

While Guu et al. (2018) uses sum of inserted/deleted word vectors as the mean direction parameter of vMF distribution $q(\mathbf{z}|\mathbf{t}, \mathbf{x})$, we choose a more powerful encoder following recent advances on

representing edits (Yin et al., 2019). Specifically, a standard diffing algorithm is run to compute an alignment of tokens in $\mathbf{t}$ and $\mathbf{x}$, and produces two aligned sequences $\mathbf{t}', \mathbf{x}'$ and an additional edit sequence that indicates edit operation $+$ (insertion), $-$ (deletion), $\leftrightarrow$ (substitution), and $=$ (equal) at each position. We use $\varnothing$ to denote padding. This is illustrated in Figure 2. Word embeddings of all three sequences are concatenated and fed into a single-layer LSTM to obtain the edit representation.

## 4 Experiments

Our experiments below are designed to (1) examine the efficacy of the proposed sparse neural editor on language modeling, (2) examine the efficiency of sparse neural editor on memory savings and speed-up at test time, and (3) demonstrate the interpretable semantics/syntax captured by prototypes and edit vectors.

### 4.1 Setup

We perform experiments on three different-scale datasets to test our method in different scenarios:

- MSCOCO (Lin et al., 2014): MSCOCO is an image caption dataset and we only focus on its captions as our data. The average length of captions is 12.6. The sentences do not have complex variations and are easy to find similar sentences as prototypes. We randomly sample 40K sentences as our training data and 4K as validation and test set respectively. This dataset represents a simple and small-scale setting to test our approach.

- Yelp Medium/Yelp Large: These datasets consist of sentences from Yelp restaurant reviews (Yelp, 2017) preprocessed by Guu et al. (2018), allowing us to perform a direct comparison with their method. The medium and large datasets consist of 1.5M and 17M sentences respectfully, allowing us to test in moderate and relatively large data settings respectively. Note that Yelp Medium is obtained by further filtering Yelp Large to keep sentences that are generally shorter and have less variations, but the test sets for these two are the same.

**Baselines:** We mainly consider neural language models (NLM) and the neural editor (Guu et al., 2018) as our baseline. It is worth noting that the neural editor model does not have likelihood defined on test sentences that are not similar to any example in the prototype library, thus it is necessary to interpolate with another NLM at test time for smoothing purposes, while our model is able to be used on its own. Note that we only report the neural editor baseline on Yelp Large since their public code is not ready to run on other datasets due to the required pre-built index to retrieve prototypes.

**Evaluations:** We evaluate language modeling performance with perplexity (PPL). For our model, we approximate the log marginal data likelihood through 1000 importance-weighted latent variable samples (Burda et al., 2015) and compute PPL based on this likelihood. At test time our approach prunes and has access to only $M$ prototypes that occupy 90% probability mass of the posterior prototype distribution as described in Section 3.2. We report $M$ as "#prototypes". To have an intuitive notion about how similar the prototype and data examples are, we compute average of smoothed sentence-level BLEU scores (Lin & Och, 2004) of the data examples on the validation set with their most likely prototype as the reference. We also report BLEU scores based on part-of-speech sequences (POS-BLEU)[5] to view the similarity from a more syntactic perspective. Test speed is evaluated on a single Nvidia 1080 Ti GPU.

**Hyperparameters:** We try different Dirichlet prior parameters $\alpha$ to control the sparsity and report different sparsity settings for all the datasets. The temperature parameter $\mu$ in the prototype retriever $q(\mathbf{t}|\mathbf{x})$ is tuned on the MSCOCO validation data and set as 0.3 for all datasets. The concentration parameter $\kappa$ of the vMF distribution is tuned and set as 30 for MSCOCO and Yelp Medium and 40 for Yelp Large. The number of Reinforce samples $L$ is set as 10 across all datasets. We sample 50K examples for Yelp Medium and 100K examples for Yelp Large as our training prototype library to address the training memory issue discussed in Section 3.2. We employ the same attentional LSTM Seq2Seq model as in Guu et al. (2018) to parameterize $p_{\boldsymbol{\gamma}}(\mathbf{x}|\mathbf{t}, \mathbf{z})$ for a direct comparison. Our implementation is based on the fairseq toolkit (Ott et al., 2019) and complete hyperparameter settings can be found in Appendix B.

Table 1: Results on three datasets. Numbers in the parentheses indicate the percentage of prototypes over all training examples. BLEU score is computed by comparing validation examples against their most likely prototypes. POS-BLEU represents the BLEU score on part-of-speech sequences. We also list BLEU scores from random prototype retrieval as a reference point. Results in the starred entry (∗) are obtained by running the public code of the neural editor.

| Dataset | Model | PPL↓ | #prototypes | test speed (sent/s)↑ | BLEU | POS-BLEU |
|---|---|---|---|---|---|---|
| **MSCOCO** | random retrieval | – | – | – | 10.9 | 31.6 |
| | NLM | 20.0 | – | 3714 | – | – |
| | Sparse Neural Editor ($\alpha = 10^{-3}$) | 18.9 | 25 (0%) | 388 | 13.2 | 38.8 |
| | Sparse Neural Editor ($\alpha = 0.1$) | **18.6** | 778 (2%) | 313 | 17.2 | 42.5 |
| | Sparse Neural Editor ($\alpha = 0.2$) | 19.0 | 16K (40%) | 250 | 20.9 | 46.7 |
| | Sparse Neural Editor ($\alpha = 0.3$) | 19.2 | 22K (56%) | 217 | 22.2 | 47.9 |
| **Yelp Medium** | random retrieval | – | – | – | 8.1 | 17.8 |
| | NLM | 74.7 | – | 236 | – | – |
| | Sparse Neural Editor ($\alpha = 10^{-3}$) | 63.6 | 77 (0%) | 157 | 12.3 | 24.7 |
| | Sparse Neural Editor ($\alpha = 0.5$) | **61.9** | 1.5K (0.1%) | 107 | 21.6 | 38.4 |
| | Sparse Neural Editor ($\alpha = 10$) | 63.2 | 31K (2.1%) | 95 | 29.9 | 48.3 |
| **Yelp Large** | random retrieval | – | – | – | 6.6 | 16.0 |
| | NLM | 34.2 | – | 272 | – | – |
| | Sparse Neural Editor ($\alpha = 0.7$) | **30.2** | 2K (0.01%) | 108 | 10.5 | 24.8 |
| | Sparse Neural Editor ($\alpha = 10$) | 30.3 | 5.5K (0.03%) | 98 | 10.8 | 25.3 |
| | *Interpolated w/ NLM* | | | | | |
| | Neural Editor (Guu et al., 2018) | 26.9 | 17M (100%) | – | – | – |
| | Neural Editor (our runs)∗ | 31.2 | 17M (100%) | 0.1 | – | – |
| | Sparse Neural Editor ($\alpha = 0.7$) | **20.2** | 2K (0.01%) | 108 | 10.5 | 24.8 |

## 4.2 Results

Results are shown in Table 1. Our sparse neural editor outperforms the NLM baseline across all datasets in terms of PPL, often by a large margin. When interpolated with an NLM at test time, our method outperforms the NLM baseline by 14 PPL points and neural editor by 6.7 PPL points.[6] This effect is also observed in (Guu et al., 2018) – prototype-driven language models are especially strong at modeling test sentences that have similar prototypes but relatively weak at modeling others, thus interpolation with a normal NLM is likely to help. Furthermore, in line with the goal of this work to learn a sparse prototype set, our method is able to achieve superior language modeling performance while utilizing only a small fraction of training examples as prototypes. This verifies our hypothesis that a sparse prototype set suffices for such non-parametric language modeling. Also, sparsity learned in our model allows for over a 1000x memory savings and 1000x speed-up[7] at test time on Yelp Large compared with a previous neural editor that memorizes all training examples.

Table 1 demonstrates the trend that a smaller Dirichlet hyperparameter $\alpha$ leads to a sparser prototype set, which agrees with our expectation. BLEU scores are also improved as $\alpha$ increases, implying that the less sparse the prototype set the closer the match between the sentence and its prototype. Interestingly, the BLEU score on Yelp Large is a bit low and the model tends to generally favor sparse prototypes. We suspect that this is because it is difficult to learn prototypes that capture fine-grained semantic features and large sentence variations among 17M examples with a limited prototype memory budget, thus the model has to learn prototypes that represent more generic shared features among examples to reach an optimum – for example, the syntactic feature as somewhat reflected by the decent POS-BLEU scores.

We want to emphasize that different sparsity may lead to different notions of prototypes and it is hard to judge which one should be preferred – memorizing more prototypes pays cost on language modeling performance and does not necessarily produce better PPL. Also, prototypes that are "less similar" to the examples on the superficial level but capture coarse-grained features may have potentially interesting application on sentence generation since the model is able to generate more diverse output conditioned on prototypes.

Table 2: Number of matching tokens between examples and their prototypes on the Yelp Medium validation set. Results are reported in cluster of POS tags. Relative changes that are larger than the overall change are bolded.

| Model | Overall | NOUN | DET | AUX | PRON | ADJ | VERB | CCONJ |
|---|---|---|---|---|---|---|---|---|
| Sparse Neural Editor (31K prototypes) | 91.2K | 14.4K | 9.6K | 9.3K | 9.0K | 7.2K | 6.4K | 5.5K |
| Sparse Neural Editor (1.5K prototypes) | 74.7K | 9.9K | 8.5K | 8.2K | 7.3K | 5.6K | 4.4K | 5.0K |
| Relative Change | -18.1% | **-31.3**% | -11.5% | -11.8% | **-18.9**% | **-22.2**% | **-31.3**% | -9.1% |

Table 3: Qualitative examples of prototypes when using denser and sparser prototype supports.

| Data Examples | Prototypes |
|---|---|
| the best corned beef hash i 've ever had ! | (dense) the best real corned beef hash i 've had . <br> (sparse) the chicken satay is the best i 've ever had . |
| the grilled chicken was flavorful , but too flavorful . | (dense) the chicken was moist but it lacked flavor . <br> (sparse) my sandwich was good but the chicken was a little plain . |
| i asked her what time they close and she said \<cardinal\> o'clock . | (dense) i asked what time they closed \<date\> , and was told \<cardinal\> . <br> (sparse) we asked how long the wait was and we were informed it would be \<time\> . |

Table 4: Qualitative examples from the MSCOCO dataset on interpolated sentence generation given the prototype. The first row is the given prototype, the second-row and the last-row sentences are obtained by sampling edit vectors from the prior, the rest three sentences are generated by interpolating between the two edit vectors.

| Prototype: A man is using a small laptop computer | Prototype: A cat sitting on a sidewalk behind a bush |
|---|---|
| A man is using his laptop computer with his hands on the keyboard | A cat laying on top of a wooden bench |
| A man is using a laptop computer with his hands on the keyboard | A cat standing next to a tree in a park |
| A man is using a laptop computer while sitting on a bench | Two cats sitting on a bench near a park bench |
| A man is using a laptop computer in the middle of a room | A dog sitting on a bench near a park bench |
| A young man is using a laptop computer in the middle of a room | A dog sitting on a bench near a park bench |

## 4.3 Analysis

**How do prototypes change when they grow sparser?** In Table 1 we notice the BLEU scores usually drop when the prototype set is sparser, implying the learned prototypes change in some way under different sparsity settings. Here we take the Yelp Medium dataset as an example to analyze how prototypes are trained to capture sentence attributes differently in the sparse ($\alpha = 0.5$) and relatively dense ($\alpha = 10$) situations. Specifically, we align prototype and example sequences to minimize edit distance, and focus on words that were aligned between the two sequences. This allows us to obtain a notion of what kind of words are more likely to be kept the same as the prototype during the model's editing process and how this pattern changes in different sparsity settings. We cluster these matched words in terms of POS tag and report the most common ones.[8]

Results are shown in Table 2. While the overall number of matching tokens decreases as the prototype set becomes sparser, the content words exhibit a more drastic change (e.g. the nouns, adjectives, and verbs). In contrast, the function words experience a moderate decrease only (e.g. the determiners, auxiliaries, and coordinating conjunctions). This shows that the model tends to learn prototypes that drop fine-grained semantic distinctions but keep the same general syntax when a sparsity constraint is enforced, which is not surprising since it is difficult for a limited number of prototypes to capture large semantic variations in a large dataset. We list qualitative examples in Table 3 to demonstrate this phenomenon, where some content words such as "beef" or "chicken" differ between data examples and prototypes in the sparser setting, yet these words do not change in the dense setting.

**Ablation Study on Pre-clustering Prototypes:** There is a simple method that is able to endow the neural editor baseline with sparse prototypes – pre-clustering the training examples and using only a subset of the them as the prototype pool during training and test. To compare the effectiveness of this heuristic sparse prototypes and our learned sparse prototypes, we experiment this baseline on MSCOCO dataset and pre-cluster 778 prototypes that correspond to our setting $\alpha = 0.1$. Concretely, we run k-means to obtain 778 clusters of training sentences embedded by Sentence-BERT (Reimers & Gurevych, 2019) which produces state-of-the-art semantic sentence embeddings. Then for each cluster we identify one prototype whose embedding is the closest to the cluster centroid. We use these 778 prototypes in the neural editor model (Guu et al., 2018) and train with the same objective

Table 5: Comparison with Neural Editor that uses heuristic pre-clustering to select a sparse prototype support. The results are on MSCOCO dataset.

| Model | PPL↓ | #prototypes | BLEU | POS-BLEU |
|---|---|---|---|---|
| NLM | 20.0 | – | – | – |
| Neural Editor (pre-cluster) | 19.5 | 778 | 17.9 | 40.4 |
| Sparse Neural Editor | 18.6 | 778 | 17.2 | 42.5 |

as Guu et al. (2018).[9] Results are shown in Table 5. Our method outperforms the neural editor with the same number of prototypes and presents similar BLEU scores, which demonstrates the superiority of learned prototypes over the heuristically selected prototypes.

**Interpolation on the edit vector space:**    We take our model with 1.5K prototypes on the MSCOCO dataset (i.e. $\alpha = 0.1$) and perform sentence interpolation in edit vector space. Specifically, we sample two edit vectors from the uniform vMF prior to produce two sentences for each prototype with beam search decoding, then we perform spherical linear interpolation of the two edit vectors to generate interpolated sentences in-between. Qualitative examples in Table 4 (more examples in Appendix C) show that the edit vectors are able to *smoothly* capture minor edits over the given prototypes.

## 5    Conclusion

In this work, we propose a novel generative model that discovers a sparse prototype set automatically by optimizing a variational lower bound of the log marginal data likelihood. We demonstrate its effectiveness on language modeling and its efficiency advantages over previous prototype-driven generative models. The framework proposed here might be generalized to automatically discover salient prototypes from a large corpus. New kinds of prototype structure in text might be discovered through either injecting different biases into the model (e.g. sparsity biases in this paper), or incorporating prior knowledge into the prototype library before training. Finally, the approach might be easily extended to conditional generation (e.g. with the edit vectors depending on other data input), and we envision that inducing a sparse prototype set in this case may potentially facilitate controlling text generation through prototypes. We leave exploration in this direction as our future work.

## Broader Impact

Our approach is likely to benefit researchers or practitioners who are interested in generating text through machines in practical scenarios such as writing news articles given facts, describing stock trends with stock data, generating headlines, etc. Many such applications have a small number of templates for generated text. On the one hand, our model is able to automatically induce those representative prototypes from a large corpus, helping knowing the salient prototypes or templates to help human writers to start with. On the other hand, our approach can be easily extended to these conditional text generation task directly, for example, with the edit vector depending on the input data instead of from a uniform distribution. Such way potentially allows our model to control the prototypes and directly generate text conditioned on the input data as well. Furthermore, the prototype library may be further explored in other formats in addition to training examples, opening a door to more flexible control under different notions of "prototypes".

With respect to possible disadvantages from this research from a societal perspective, as a contribution to the widely studied field of language modeling, the proposed method inherits some of the risks of the field as a whole. These may include models being used maliciously to create fake content (Zellers et al., 2019), or models being used in earnest being manipulated through adversarial attacks to generate undesirable or defamatory content (Wallace et al., 2019). With respect to the latter, as our method is more interpretable due to its use of readable prototypes, we actually expect that it may be *more* robust to adversarial attacks, and these attacks may be easier for human auditors to detect or defuse when they occur. However, this is speculation, and would have to be confirmed by further experiments.

## Acknowledgements

This work was supported in part by the DARPA GAILA project (award HR00111990063), and a gift of computation credits from Amazon AWS. The views and conclusions contained in this document are those of the authors and should not be interpreted as representing the official policies, either expressed or implied, of the U.S. Government or Amazon.

## Footnotes

[1]Code is available at https://github.com/jxhe/sparse-text-prototype.

[2]Below, we sometimes ignore the subscript to simplify notation when there is no confusion.

[3]$q_{\boldsymbol{\lambda}}(\theta)$ is not symmetric and $\boldsymbol{\lambda}$ is a vector.

[4]We use pretrained uncased BERT base from the transformers library (Wolf et al., 2019).

[5]POS tagging is performed using the Stanza library (Qi et al., 2020).

[6] For a fair comparison, we interpolate with the same pretrained NLM from (Guu et al., 2018).

[7] We include the time to retrieve prototypes for new test sentences when computing test speed. We use Guu et al. (2018)'s public implementation of the neural editor, where the computation of edit distance between all training examples and test sentences to find nearest neighbors accounts for much of the runtime. More efficient implementation of this operation, for example through tries, may speed this to some extent.

[8]Note that alignment is performed on word sequences instead of POS sequences.

[9]The prior over edit vector $p(\mathbf{z})$ and the inverse neural editor $q(\mathbf{z}|\mathbf{t}, \mathbf{x})$ in sparse neural editor is different from the original neural editor model. For a fair comparison we use $p(\mathbf{z})$ and $q(\mathbf{z}|\mathbf{t}, \mathbf{x})$ described in this paper for the neural editor baseline here.

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
