[Supplementary Material]

# A  Derivations of Variational Inference and ELBO

## A.1  Derivation of optimal $q^*(\theta)$

Here we try to show that the optimal variational distribution over $\theta$, $q^*_{\boldsymbol{\lambda}}(\theta)$, is a Dirichlet distribution and derive its optimal $\boldsymbol{\lambda}^*$ as given in Eq. 7. According to (Bishop, 2006) (Chapter 10.1.1), we have:

$$q^*_{\boldsymbol{\lambda}}(\theta) \propto \exp\left(\mathbb{E}_{-\theta}[\log p(\{\mathbf{x}_n, \mathbf{t}_n, \mathbf{z}_n\}_{n=1}^N, \theta)]\right), \tag{10}$$

where $\mathbb{E}_{-\theta}$ denotes expectation over all latent variables except for $\theta$. We expand Eq. 10 as:

$$
\begin{aligned}
q^*_{\boldsymbol{\lambda}}(\theta) &\propto \exp\left(\mathbb{E}_{-\theta}[\log p(\{\mathbf{x}_n, \mathbf{t}_n, \mathbf{z}_n\}_{n=1}^N, \theta)]\right) \\
&\propto \exp\left(\mathbb{E}_{-\theta}[\log p_\alpha(\theta) + \sum_n \log p(\mathbf{t}_n|\theta)]\right) \\
&\propto \exp\left(\mathbb{E}_{-\theta}[\sum_k \log \theta_k^{\alpha-1} + \sum_n \log \prod_k \theta_k^{\mathbb{1}(\mathbf{t}_n=\mathbf{x}_k)}]\right) \\
&\propto \exp\left(\mathbb{E}_{-\theta}[\sum_k \log \theta_k^{\alpha-1} + \sum_n \sum_k \mathbb{1}(\mathbf{t}_n=\mathbf{x}_k)\log \theta_k]\right) \\
&\propto \exp\left(\sum_k \log \theta_k^{\alpha-1} + \sum_n \sum_k q(\mathbf{t}_n=\mathbf{x}_k|\mathbf{x}_n)\log \theta_k\right) \\
&\propto \prod_k \theta_k^{\alpha-1+\sum_n q(\mathbf{t}_n=\mathbf{x}_k|\mathbf{x}_n)},
\end{aligned}
\tag{11}
$$

where $\mathbb{1}(\cdot)$ is the indicator function. We conclude that $q^*_{\boldsymbol{\lambda}}(\theta)$ has the form of Dirichlet distribution and the optimal Dirichlet parameter $\lambda_k^* = \alpha + \sum_n q(\mathbf{t}_n = \mathbf{x}_k|\mathbf{x}_n)$.

## A.2  Derivation of Three KL Divergence Terms

There are three KL divergence terms in our training objective ELBO (Eq. 4). Now we show that all three KL divergence terms can be computed exactly and efficiently at training time and we derive their expressions respectively:

**(1).** $\mathbb{E}_{q(\mathbf{t}_n|\mathbf{x}_n)}[D_{\mathrm{KL}}(q(\mathbf{z}_n|\mathbf{t}_n,\mathbf{x}_n)||p(\mathbf{z}_n))]$**:**  As shown in (Xu & Durrett, 2018), the KL divergence between any vMF distribution with fixed concentration parameter and a uniform vMF distribution is a constant:

$$
\begin{aligned}
D_{\mathrm{KL}}(\mathrm{vMF}(\mu,\kappa)||\mathrm{vMF}(\cdot,0)) = {}& \kappa\frac{I_{d/2}(\kappa)}{I_{d/2-1}(\kappa)} + (\frac{d}{2}-1)\log\kappa - \frac{d}{2}\log(2\pi) \\
& - \log I_{d/2-1}(\kappa) + \frac{d}{2}\log\pi + \log 2 - \log\Gamma(\frac{d}{2}),
\end{aligned}
\tag{12}
$$

where $d$ is the number of dimensions, $I_v$ stands for the modified Bessel function of the first kind at order $v$. Therefore,

$$\mathbb{E}_{q(\mathbf{t}_n|\mathbf{x}_n)}[D_{\mathrm{KL}}(q(\mathbf{z}_n|\mathbf{t}_n,\mathbf{x}_n)||p(\mathbf{z}_n))] = D_{\mathrm{KL}}(\mathrm{vMF}(\mu,\kappa)||\mathrm{vMF}(\cdot,0)) = const \tag{13}$$

**(2).** $\mathbb{E}_{q_{\boldsymbol{\lambda}}(\theta)}[D_{\mathrm{KL}}(q(\mathbf{t}_n|\mathbf{x}_n)||p(\mathbf{t}_n|\theta))]$**:**

$$
\begin{aligned}
& \mathbb{E}_{q_{\boldsymbol{\lambda}}(\theta)}[D_{\mathrm{KL}}(q(\mathbf{t}_n|\mathbf{x}_n)||p(\mathbf{t}_n|\theta))] \\
={}& \mathbb{E}_{q_{\boldsymbol{\lambda}}(\theta)}\left[\mathbb{E}_{q(\mathbf{t}_n|\mathbf{x}_n)}\left[\log q(\mathbf{t}_n|\mathbf{x}_n) - \log p(\mathbf{t}_n|\theta)\right]\right] \\
={}& \mathbb{E}_{q(\mathbf{t}_n|\mathbf{x}_n)}\log q(\mathbf{t}_n|\mathbf{x}_n) - \mathbb{E}_{q_{\boldsymbol{\lambda}}(\theta)}\mathbb{E}_{q(\mathbf{t}_n|\mathbf{x}_n)}\left[\log\prod_k \theta_k^{\mathbb{1}(\mathbf{t}_n=\mathbf{x}_k)}\right] \\
={}& \sum_k q(\mathbf{t}_n|\mathbf{x}_n)\log q(\mathbf{t}_n|\mathbf{x}_n) - \mathbb{E}_{q_{\boldsymbol{\lambda}}(\theta)}\mathbb{E}_{q(\mathbf{t}_n|\mathbf{x}_n)}\left[\sum_k \mathbb{1}(\mathbf{t}_n=\mathbf{x}_k)\log\theta_k\right] \\
={}& \sum_k q(\mathbf{t}_n=\mathbf{x}_k|\mathbf{x}_n)\log q(\mathbf{t}_n=\mathbf{x}_k|\mathbf{x}_n) - \sum_k q(\mathbf{t}_n=\mathbf{x}_k|\mathbf{x}_n)\mathbb{E}_{q_{\boldsymbol{\lambda}}(\theta)}\log\theta_k \\
\overset{(i)}{=}{}& \sum_k q(\mathbf{t}_n=\mathbf{x}_k|\mathbf{x}_n)\log q(\mathbf{t}_n=\mathbf{x}_k|\mathbf{x}_n) - \sum_k q(\mathbf{t}_n=\mathbf{x}_k|\mathbf{x}_n)[\Psi(\lambda_k) - \Psi(\sum_i^N \lambda_i)],
\end{aligned}
\tag{14}
$$

where $\Psi(\cdot)$ is the digamma function, and step $(i)$ computes the expectation of $\log\theta_k$ over Dirichlet variable $\theta$ by using the general fact that the derivative of the log normalization factor with respect to the natural parameter is equal to the expectation of the sufficient statistic.

**(3).** $D_{\mathrm{KL}}(q_{\boldsymbol{\lambda}}(\theta)||p_{\alpha}(\theta))$**:**

$$
\begin{aligned}
D_{\mathrm{KL}}(q_{\boldsymbol{\lambda}}(\theta)||p_{\alpha}(\theta)) &= \mathbb{E}_{q_{\boldsymbol{\lambda}}(\theta)}\big[\log q_{\boldsymbol{\lambda}}(\theta) - \log p_{\alpha}(\theta)\big] \\
&= \mathbb{E}_{q_{\boldsymbol{\lambda}}(\theta)}\big[\log\big(\prod_k \theta_k^{\lambda_k-1}/B(\boldsymbol{\lambda})\big) - \log\big(\prod_k \theta_k^{\alpha-1}/B(\alpha)\big)\big] \\
&= -\log B(\boldsymbol{\lambda}) + \sum_k (\lambda_k - 1)[\Psi(\lambda_k) - \Psi(\sum_i^N \lambda_i)] \\
&\quad + \log B(\alpha) - \sum_k (\alpha - 1)[\Psi(\lambda_k) - \Psi(\sum_i^N \lambda_i)],
\end{aligned}
\tag{15}
$$

where $B(\cdot)$ is the multivariate beta function and $B(\boldsymbol{\lambda})$ is the normalization factor for Dirichlet distribution parameterized by $\boldsymbol{\lambda}$.

# B  Experimental Details

On MSCOCO dataset, we use a single-layer attentional LSTM seq2seq architecture with word embedding size 100 and hidden state size 400 as $p_{\boldsymbol{\gamma}}(\mathbf{x}|\mathbf{t},\mathbf{z})$, the latent edit vector dimension is 50. This configuration follows the hyperparameters for vMF-VAE (Xu & Durrett, 2018). On Yelp Medium and Yelp Large datasets, we follow (Guu et al., 2018) to use a three-layer attentional LSTM seq2seq architecture as $p_{\boldsymbol{\gamma}}(\mathbf{x}|\mathbf{t},\mathbf{z})$ with word embedding size 300 and hidden state size 256, the edit vector dimension is 128. Skip connections are also used between adjacent LSTM layers. In the inverse editor $q_{\boldsymbol{\phi}_{z|t,x}}(\mathbf{z}|\mathbf{t},\mathbf{x})$, we use a single-layer LSTM to encode three sequences – the aligned prototype, aligned data example, and the edit operation sequence, of which the word embedding size for text sequences and the hidden state size are the same as in $p_{\boldsymbol{\gamma}}(\mathbf{x}|\mathbf{t},\mathbf{z})$, and the word embedding size for edit operation sequence is 10 (since the vocabulary size of edit operations is very small). Across all datasets, we initialize word embeddings in $p_{\boldsymbol{\gamma}}(\mathbf{x}|\mathbf{t},\mathbf{z})$ (both encoder and decoder sides) and $q_{\boldsymbol{\phi}_{z|t,x}}(\mathbf{z}|\mathbf{t},\mathbf{x})$ with GloVe word embeddings (Pennington et al., 2014) following (Guu et al., 2018). All NLM baselines use the same architecture as $p_{\boldsymbol{\gamma}}(\mathbf{x}|\mathbf{t},\mathbf{z})$ in our model for a fair comparison.

With respect to hyperparameter tuning, we tune the temperature parameter $\mu$ in the prototype retriver $q(\mathbf{t}|\mathbf{x})$ on the MSCOCO validation data in the range of $\{0.1, 0.3, 0.5, 0.7, 0.9, 1.0\}$, and set as 0.3 for all datasets. The concentratioon parameter $\kappa$ of the vMF distribution is tuned in the range of $\{30, 40, 50\}$ for all datasets. $\kappa$ is set as 30 for MSCOCO and Yelp Medium and 40 for Yelp Large. We run different $\alpha$ as $\{0.1, 0.3, 0.5, 0.7, 0.9, 1, 10\}$ for each dataset to obtain prototype set with varying sparsity. On MSCOCO dataset we also add an additional run with $\alpha = 0.2$ since 0.5 is a large value on this dataset already (90% of training examples are selected as the prototype set when $\alpha = 0.5$). We apply annealing and free-bits techniques following (Li et al., 2019) to the KL term on prototype variable, $\mathbb{E}_{q_{\boldsymbol{\lambda}}(\theta)}[D_{\mathrm{KL}}(q(\mathbf{t}_n|\mathbf{x}_n)||p(\mathbf{t}_n|\theta))]$, to mitigate posterior collapse. Specifically, $\mathbb{E}_{q_{\boldsymbol{\lambda}}(\theta)}[D_{\mathrm{KL}}(q(\mathbf{t}_n|\mathbf{x}_n)||p(\mathbf{t}_n|\theta))]$ in our training objective becomes $\beta \cdot \max\{\mathbb{E}_{q_{\boldsymbol{\lambda}}(\theta)}[D_{\mathrm{KL}}(q(\mathbf{t}_n|\mathbf{x}_n)||p(\mathbf{t}_n|\theta))], c\}$ in practice. This objective means that we can downweight this KL term with $\beta < 1$ and optimize it only when this KL is larger than a threshold value $c$. We increase $\beta$ from 0 to 1 linearly in the first $m$ epochs (annealing). $m$ is tuned in the range of $\{5, 10\}$ for MSCOCO and $\{1, 2, 3\}$ for Yelp Medium and Yelp Large.[10] $c$ is tuned in the range of $\{5, 6, 8\}$. To obtain the reported results in Section 4, $m$ is set as 5 for MSCOCO, 2 for Yelp Medium and 3 for Yelp Large. $c$ is set as 5 for MSCOCO, 6 for Yelp Medium and 8 for Yelp Large. We use Adam (Kingma & Ba, 2014) to optimize the training objective with learning rate 0.001.

# C Qualitative Results on Interpolation

As in Section 4.3, here we show more generated examples through interpolation on MSCOCO dataset.

Table 6: Qualitative examples from the MSCOCO dataset on interpolated sentence generation given the prototype. For each example, the first row is the given prototype, the second-row and the last-row sentences are obtained by sampling edit vectors from the prior, the rest three sentences are generated by interpolating between the two edit vectors.

| Prototype: a horse drawn carriage on the side of a city street | Prototype: A baseball pitcher on the mound having just threw a pitch |
| --- | --- |
| Two horses drawn carriage on a city street | a baseball player swinging a bat at home plate |
| Two horses standing next to each other on a city street | a man about to hit a ball with his bat |
| Two horses on the side of a city street | a man swinging a bat at the ball during a game |
| Two horses on the side of a city street | a person swinging a bat at the ball during a game |
| A brown and white horse drawn carriage on a city street | A person swinging a bat during a baseball game |
| **Prototype: A man walking on the beach carrying a surfboard** | **Prototype: A group of people are raising an umbrella on a beach** |
| Two people standing next to each other on a beach | A group of people are walking on the beach with umbrellas |
| A person standing on the beach holding a surfboard | A group of people are walking on the beach next to each other |
| A man walking along the beach with a surfboard | A group of people are walking on the beach with umbrellas |
| A man walking on the beach with a surfboard | A group of people are holding umbrellas on the beach |
| A young man walking on the beach with a surfboard | A group of people are walking on the beach |
| **Prototype: there is a white truck that is driving on the road** | **Prototype: A couple of bags of luggage sitting up against a wall** |
| there are many cows that are standing in the dirt | A large pile of luggage sitting on top of a wall |
| there are many cows that are standing in the dirt | A pile of luggage sitting on top of a wall |
| the truck is driving down the road in the rain | Two bags of luggage sitting on the ground |
| this truck is driving down the road in the rain | Two bags of luggage sitting in a room |
| This truck is pulled up to the side of the road | A couple of bags of luggage on a wooden floor |
| **Prototype: A man riding a sailboat in the ocean next to a shore** | **Prototype: A beer bottle sitting on a bathroom sink next to a mirror** |
| A man on a boat in a body of water | A white cell phone sitting next to a toilet in a bathroom |
| A man riding a boat on a body of water | A white bottle of wine sitting next to a toilet |
| A man riding a boat in a body of water | A glass of wine sitting next to a toilet in a bathroom |
| A man riding a small boat on a body of water | A pair of scissors is placed next to a toilet |
| A man riding a wave on top of a boat | A pair of scissors sitting next to each other on a toilet |
| **Prototype: A little boy sitting on a mattress holding a stuffed animal** | **Prototype: A giraffe has its nose pressed against the trunk of a tree** |
| A little girl playing with a stuffed animal | Two giraffes look at a wire fence to eat |
| A little girl playing with a stuffed animal | Two giraffes look at a fence to eat |
| A little boy holding a stuffed animal in his mouth | a couple of giraffes are standing by a fence |
| A little girl sitting on a bed with stuffed animals | a close up of a giraffe is eating a carrot |
| A little girl sitting on a bed with stuffed animals | a close up of a giraffe has its mouth open |

## Footnotes

[10]It is unreasonable to set a large $m$ on Yelp dataset since there are tens of thousands update steps per epoch on Yelp, the annealinig process would be too slow if $m$ is large which usually hurts the language modeling performance (He et al., 2019).