[Reviews · NeurIPS 2020]

Review 1

Summary and Contributions: This paper is largely an upgraded version of Guu et al. (2018). Two major changes are made: first, modeling a sparse distribution over prototypes with a Dirichlet prior over a multinomial, and second, actually learning this sparse distribution. The paper uses amortized variational inference, further approximating the gradients using REINFORCE to deal with the large number of prototypes. The variational approximation for prototypes t ends up being a multinomial over prototypes, and q(z|t,x) is modeled with a vMF distribution representing an edit vector, similar to the editor p(x|t,z) in the forward model. Results show lower perplexity than the neural editor of Guu et al. Having more templates leads to higher BLEU score matches with prototypes (somewhat unsurprisingly).

Strengths: This paper is very clear, presents a clean idea, and has experiments to back that idea up. Actually modeling the prototype choice addresses a main shortcoming of the original Guu et al. work, and it feels like the paper's modeling approaches and variational inference are naturally fit to the problem and as simple as they can be given the complexity of the setting. The results are solid and show improvements over both the model of Guu et al. and also over basic NLMs. Perplexity is lower with sparser prototype sets (I'm assuming because the p(t|theta) term is less costly) but BLEU score is also lower, suggesting a lower quality match. From Table 2, it appears that these prototypes are behaving as advertised, with content words being edited.

Weaknesses: This is a fairly nice paper overall, but it feels a bit thin on major new ideas or impactful experiments. It takes the model of Guu et al and upgrades it in terms of sophistication, achieving better results. The model and inference techniques are certainly novel, although they follow a well-trodden path of techniques like VAEs using variational inference. But that approach feels limited: while the paper references some nice instances where this problem of prototype generation shows up, I'm not convinced that this approach works all that well in general. Yelp and MSCOCO are large amounts of text in relatively limited domains, and I'm curious whether this really scales to corpora like newswire or web text which feature more diverse sentence structures and topics. I like the motivation the authors give in the Broader Impacts section. I'm willing to suspend my skepticism and advocate for this paper in the hope that these techniques can prove useful to other practitioners. Certainly this might be a path towards reducing the crazy amount of overparameterization in current-generation language models. But perplexity on these existing datasets doesn't feel like the right direction to be working towards to prove out these techniques. I think in order to be excited about a *next* paper in this modeling space, I'd need to see some stronger evidence that this is a good direction to pursue.

Correctness: Yes, the paper is technically solid and the claims and experiments seem correct.

Clarity: Figure 1 is very nice and the paper is presented very clearly.

Relation to Prior Work: Yes

Reproducibility: Yes

Additional Feedback: ======= Post-response: interesting response, especially the pre-clustering experiments. The rest of my review is unchanged.


Review 2

Summary and Contributions: This paper proposes an approach to learning a sparse prototype distribution for use in a select-prototype-and-then-edit style language model. Rather than assume a uniform distribution over prototypes (as in previous work), the authors view a global distribution over prototypes as a possibly sparse latent variable. This latent is learned with an SVI-style update, while the other latents in the model are learned with amortized variational inference. The authors demonstrate improved perplexity against the original neural editor model, as well as faster and more memory efficient at test time.

Strengths: - The paper is well motivated and timely; as the authors note, there appears to be increased interest in augmenting language models with retrieved data. - The approach is reasonable and appears to obtain good results. - The paper is written clearly and is generally easy to follow.

Weaknesses: - I don't see any significant weaknesses, though I have some minor concerns about baselines; see below.

Correctness: The claims and methodology appear correct. Some minor concerns: - It seems that a natural baseline which avoids learning theta would be to simply pre-cluster the sentences in the training set (using a number of centroids equal to the desired size of the prototype set), and then learn the model assuming a uniform prior over these centroids. I think the proposed approach is certainly more elegant than this, but it would be useful to know how much improvement (if any) you obtain over this naive approach. - It looks like the original Guu et al. PPL results use a looser lower bound on the marginal, using only a single latent variable sample, as opposed to your 1000 importance sample approximation. If their results improve with a tighter bound these should probably be reported.

Clarity: The paper is generally clear and easy to follow.

Relation to Prior Work: The paper mentions the most important related work, as far as I can tell, though it does not have an expansive related work section.

Reproducibility: Yes

Additional Feedback: Questions: - My understanding from lines 165-166 is that at test time you set the prior over t to be the truncated mean of q_{\lambda}, and this is used to compute a lower bound on the PPL (which seems fine). Is that right? - While there is a bit of discussion of this in the paper, I'm still somewhat surprised that a prototype-style model would decrease PPL so much. In particular, even if we have a very repetitive corpus, I would think that would just make it easier for a standard LM to model. Do you have a sense of what exactly is going on? It might be useful/interesting to see what sorts of tokens the prototype-based-LM is more confident about than a standard LM. --- Update after author response: thanks for responding to questions and for the pre-clustering experiment results! I think these results certainly strengthen the case for this approach.


Review 3

Summary and Contributions: This paper contributes a deep latent language model where generation is performed by editing a latent prototype. A prototype is a segment of text observed in the training data. The generator is a neural network and parameter estimation is performed via variational inference. The paper contributes a number of changes to a previous design (by Guu et al, 2018) that altogether lead to improvements in 2 benchmarks (3 variants). The most important change is a symmetric (sparse) Dirichlet prior over the parameters of the retrieval component (a categorical distribution over training instances). This leads to most mass being captured by a small subset of training instances which the authors argue enable the use of larger datasets.

Strengths: * The model design is interesting and well motivated. * Retrieving (rather than generating) prototypes awards a degree of interpretability to the model. * SVI for the variational Dirichlet posterior is quite neat. * Empirical improvements seem substantial.

Weaknesses: * Sparsity seems to affect performance at test time only, scaling to large training datasets is done by a heuristic subsampling step (which is presumably suboptimal for large datasets), addressing this (which is an arguably interesting problem) is suggested as future work. * The impact of this sub-sampling on performance was not indicated. * Sparsity may be leading to improvements even where no subsampling is needed, but that is unclear from the results presented in the paper (for example, this could be the case in MSCOCO and Yelp medium, but for MSCOCO alphas are all below 1, and for Yelp medium authors only report two seemly arbitrary values). * Qualitative examples in Tables 3 and 4 did not tell me much, perhaps the authors could walk readers through the patterns they seem to recognise.

Correctness: I could not identify major problems, but I have a few remarks. 1. Line 129, what the authors identify as a Multinomial distribution really is a Categorical distribution. A multinomial gives support to N-dimensional count vectors, and the authors seem to want a category in {1, ..., N}. Also a multinomial takes an additional parameter, the number of trials, which is not there (and indeed would not be relevant to this application) 2. Line 110 the authors claim that picking a variational family (vMF in this case) that cannot recover the prior (this is the case for vMF kappa > 0) has the "advantage of being free of the issue of posterior collapse". I cannot see why this would be true. While this mismatch leads to a lowerbound on KL(q(z|x) || p(z)), this KL is only an upperbound on I(X;Z), thus KL(q(z|x) || p(z)) > 0 is no guarantee that I(X;Z) > 0. If the authors think this is true, I'd like to ask that they point to relevant literature. Note that later in the paper free bits is used to alleviate posterior collapse.

Clarity: I found the paper mostly well-written, except for a few bits were a somewhat informal language led to less accurate statements or claims (such as the language around posterior collapse and the merits of fixed concentration in vMF).

Relation to Prior Work: No red flags that I could spot. One remark, in different places the paper comments on differences compared to the approach of Guu et al. It would have been nice to see a paragraph that gathers the crucial remarks.

Reproducibility: Yes

Additional Feedback: I enjoyed reading your manuscript, and I find the idea of generating sentences by editing prototypes an exciting direction. In the following I'd like to raise a few points, some are comments, some are clarification questions. 1. If the support of the retrieval component q(t|x) is an index set of the training data (as in, a sample t indexes a training instance), you cannot easily change the support at test time, can you? If I am right that you cannot, then you are limited to using the same repository of prototypes during training and test. This in turn means that subsampling the training data, for scalability, also affects the set of prototypes available for test-time generations. Am I correct? Then the "speedup" at test time comes from concentrating on a small subset of the potentially sub-sampled training set, is that correct? 2. In line 240, it's a bit unclear what it's meant by "interpolated with NLM at test time" (what is the exact strategy? is there an "interpolation coefficient" and what value does it take?). 3. Please acknowledge Davidson et al (2018) "Hyperspherical Variational Auto-Encoders" for hyperspherical VAEs, a differentiable reparameterisation of vMFs, and for a differentiable and numerically stable KL. -- Feedback on broader impact: if I am correct, a general caveat of sparsity is that truncation (based on most likely outcomes) may overly emphasize salient features and that, in turn, exacerbate biases in the dataset. If the authors agree with me, please consider adding a couple of remarks to the broader impact statement. — Thanks for the feedback. Currently the paper is somewhat a technical contribution (sparsity in a retrieve-and-edit approach to LM) and somewhat an attempt to push an alternative agenda to big/bigger/enormous (think GPT). I think this made the contributions in each direction look a bit thin. To strengthen the former, I’d recommend investigating some of the technical knobs more thoroughly: e.g., prior concentration, impact of the subsampling step (or alternatives to it). To strengthen the latter, I’d look into addressing R1’s comment about datasets with diverse/richer domains.


Review 4

Summary and Contributions: The paper speeds up the prototype-driven text generation system by Guu et al. During training, they add a (sparse) Dirichlet prior to all training examples (which are used to retrieve templates for generation) to encourage the generation model to rely on only a few training examples. At inference time, they can keep fewer training examples in memory by filtering only those whose posterior probability is larger than a threshold. Thus both the memory required to store training examples and the time spent on retrieving training examples is reduced. ========================== Thanks for the clarification on memory efficiency.

Strengths: The paper is excellently executed. The approach is explained with clarity. The experiments are extensive and I appreciate the authors' detailed analysis of the proposed approach, e.g. effects of the sparsity level on generated sentences.

Weaknesses: The proposed approach seems to take a detour to solve the efficiency problem by putting the retrieval system into training loop. From a practical perspective, the problem is the same as the efficiency problem in kNN models, which is well studied. Retrieval speed should be improved easily by replacing edit distance with any other inner-product based metrics. Even along the line of the proposed approach, Guu et al's model can be made faster by clustering examples and only keeping a representative one in each cluster. Obviously this method is not as principled as the one proposed, but it's simple enough to achieve the goal. On the other hand, the proposed approach has to work around the problem of limited GPU memory, since it needs all training examples in the GPU memory to update the 'retriever'. From the modeling perspective, sparsity may not be needed for good performance. Even though there are multiple potential templates in the training set, at inference time we only retrieve one for generation. In addition, pruning prototypes may lead to low performance for test examples that match uncommon prototypes. Generally, restricting the size of training data seems to go against the spirit of non-parametric modeling.

Correctness: The method looks correct to me, although I didn't check all the derivations carefully.

Clarity: The paper is very well written.

Relation to Prior Work: Yes.

Reproducibility: Yes

Additional Feedback: In table 1: why does the proposed approach work better than neural editor in terms of PPL? There are three possible sources of improvement: 1) sparse set of templates; 2) embedding-based distance instead of edit distance; 3) different representations of edit operations. I'd like to see some discussion (if not ablation study) on which one has the biggest impact. Also in table 1, with reduced size of prototypes, the BLEU score of retrieve templates goes down, which makes sense, but why does the perplexity also goes down?

[Author Response · NeurIPS 2020]

We thank the reviewers for their time and comments. Due to space limitations we could only address major points, but we'll try to reflect all advice in future revisions.

## Response to Reviewer #1

Thank you for the advice and comments! In the next revisions we are planning to add other more diverse datasets to present stronger experimental evidence.

## Response to Reviewer #2

**Q1. Pre-clustering baseline.** We have run a pre-clustering baseline on MSCOCO to respond to your concern. Specifically, we run k-means to obtain 778 clusters (corresponding to our setting $\alpha = 0.1$) of training sentences embedded by Sentence-BERT [1], then for each cluster identify one prototype whose embedding is the closest to the cluster centroid. We use these 778 prototypes and a uniform prior over them, and train with the same objective as Guu et al. (note that prototypes are retrieved based on Sentence-BERT embeddings and the edit representation is the same as ours). The importance-weighting approximated PPL is **19.5 against 18.6** from our model, which shows that the sparse prototypes selected by pre-clustering is sub-optimal compared to our method. We will add these details to the paper.

[1] Nils Reimers et al. Sentence-BERT: Sentence Embeddings using Siamese BERT-Networks. EMNLP 2019

**Q2. Setting prior over $t$ to be truncated mean of $q_\lambda(\theta)$ (on training data) at test time?** Yes, it is correct.

**Q3. Explanation about PPL improvement over NLM.** We think the PPL improvement reflects that the prototype-style model provides an easier way to generate certain examples than generating from scratch. Interpolation with NLM also leads to large improvement as observed by Guu et al. as well due to a ensembling effect of combining the two models. We will try to add quantitative detailed analysis as suggested by the reviewer to understand the behaviour.

## Response to Reviewer #3

**Q1. Sparsity may be leading to improvements even where no subsampling is needed?** When we run a dense baseline on MSCOCO with $\alpha = 10$ and 35K (90%) active prototypes, PPL is 19.9 vs. 18.6 with 778 prototypes, which implies that sparsity may lead to improvement. We'll include results from more $\alpha$ settings in the next revision.

**Q2. Inaccurate statement about posterior collapse.** This is our oversight and thank you for pointing out! We meant that non-zero KL often practically leads to non-collapsed posterior as shown in VAE text modeling literature (Xu & Durrett, 2018; Li et al., 2019). We'll make sure to correct the claims properly in next revision.

**Q3. Changing the support at test time?** This is a great point. It is possible to retrieve from a new prototype set at test time using the trained sentence encoder as in Equation (9). The prior over these new prototypes can be set by either using mean of posterior $q_\lambda(\theta)$ by running inference once on the training data, or just using a uniform prior. However, there is no sparsity guarantee in this case and we may lose the efficiency advantages. Also, your understanding is correct that the "speedup" at test time comes from concentrating on a small subset of the potentially sub-sampled training set.

**Q4. Interpolation with another NLM.** We use linear interpolation: $p(x) = \lambda p_{\text{proto}}(x) + (1 - \lambda)p_{\text{NLM}}(x)$. The coefficient $\lambda$ is 0.1 for prototype-based model, the same value as used in Guu et al. for a fair comparison.

## Response to Reviewer #4

**Q1. Other time-efficient retrieval options.** We think other methods that improve retrieval time efficiency without changing the prototype support are complementary to our approach; we aim to improve *both* memory and time efficiency by reducing prototype size. This is particularly important in the current trend towards huge training data and models.

**Q2. Pre-clustering examples?** Please see Q1 of Reviewer #2 for comparison details: sparse prototypes identified by heuristic pre-clustering are not as effective as those learned by our approach.

**Q3. Sparsity may not be needed for good performance?** We agree, but our focus is mainly on improving memory efficiency of non-parametric models at test time without losing performance. Also, the statement that "at inference time we only retrieve one for generation" is not correct – theoretically we marginalize over all latent template variables, and practically we use multiple template samples (line 219) per test example to approximate the marginal likelihood.

**Q4. Restricting prototype size goes against the spirit of non-parametric modeling?** We don't agree on this. Strictly speaking, we just encourage sparsity for the non-parametric model instead of restricting prototype size. Much information in huge training datasets is redundant, and even for traditional non-parametric kNN models that were well-studied, data reduction is one of the most important problems for work with huge data sets (kNN Wikipedia page).

**Q5. Why does our method have better PPL than neural editor?** We think the PPL improvement over the neural editor is mainly from *learnable* prototypes vs. the fixed prototypes used by Guu et al. This is further supported by our pre-clustering baseline results in Q1 of Reviewer #2.

**Q6. Why does the PPL go down with sparser prototypes?** The PPL improvement may be related to inductive bias of our model which is encouraged to focus on a sparse subset of prototypes. This might cause the decoder to find more useful ways to edit. Yet it is not always the case, for example, PPL on Yelp Medium with 77 prototypes goes up to 63.6.

[Meta-Review · NeurIPS 2020]

This paper builds upon Guu et al. (2018)’s prototype-driven text generation approach. Two major changes are made: first, modeling a sparse distribution over prototypes with a Dirichlet prior over a multinomial, and second, actually learning this sparse distribution. At training time, the paper uses amortized variational inference, further approximating the gradients using REINFORCE to deal with the large number of prototypes. At inference time, they can keep fewer training examples in memory by filtering only those whose posterior probability is larger than a threshold. Thus both the memory required to store training examples and the time spent on retrieving training examples is reduced. All four referees support accepting this paper, although two of them find it a borderline paper. The main weakness pointed out is that the proposed method is very specific to Guu et al., 2018. However, the reviewers agree this is a fairly nice contribution. The author response and the discussion clarified some concerns (mainly the point about memory efficiency raised by one of the reviewers). I agree this is a nice focused contribution, hence I recommend acceptance.